# Chronic Bacterial Infection Prevalence, Risk Factors, and Characteristics: A Bronchiectasis Population-Based Prospective Study

**DOI:** 10.3390/jcm8030315

**Published:** 2019-03-06

**Authors:** Adelina Amorim, Leonor Meira, Margarida Redondo, Manuela Ribeiro, Ricardo Castro, Márcio Rodrigues, Natália Martins, Venceslau Hespanhol

**Affiliations:** 1Pulmonology Department, Centro Hospitalar S. João, 4200-319 Porto, Portugal; lo.meira@gmail.com (L.M.); margarida.tredondo@gmail.com (M.Re.); ncmartins@med.up.pt (N.M.); hespanholv@gmail.com (V.H.); 2Faculty of Medicine, University of Porto, 4200-319 Porto, Portugal; 3Clinical Pathology Department, Centro Hospitalar S. João, 4200-319 Porto, Portugal; maria.mribeiro@hsjoao.min-saude.pt (M.Ri.); 4Radiology Department, Centro Hospitalar S. João, 4200-319 Porto, Portugal; rcastro83@sapo.pt (R.C.); marcioscrodrigues@gmail.com (M.Ro.); 5Institute for Research and Innovation in Health (i3S), University of Porto, 4200-135 Porto, Portugal

**Keywords:** bronchiectasis, chronic bacterial infection, *Pseudomonas aeruginosa*

## Abstract

Background: Few data are available on chronic bacterial infections (CBI) in bronchiectasis patients. Given that CBI seems to trigger longer hospital stays, worse outcomes, and morbimortality, this study was undertaken to assess CBI prevalence, characteristics, and risk factors in outpatients with bronchiectasis. Methods: A total of 186 patients followed in a bronchiectasis tertiary referral centre in Portugal were included. Demographic data and information on aetiology, smoking history, mMRC score, Bronchiectasis Severity Index (BSI) score, sputum characteristics, lung function, exacerbations, and radiological involvement degree were collected. Results: Patients included (mean age 54.7 ± 16.2 years; 60.8% females) were followed up for a period of 3.8 ± 1.7 years. The most common cause of bronchiectasis was infection (31.7%) followed by immune deficiencies (11.8%), whereas in 29% of cases, no cause was identified. *Haemophilus influenzae* (32.3%) and *Pseudomonas aeruginosa* (30.1%) were the most common CBI-associated possible pathogenic microorganisms. CBI patients presented a higher follow-up time than no-CBI patients (*p* = 0.003), worse lung function, BSI (*p* < 0.001), and radiological (*p* < 0.001) scores, and more prominent daily sputum production (*p* = 0.002), estimated mean volume (*p* < 0.001), and purulent sputum (*p* < 0.001). The number of exacerbations/year (*p* = 0.001), including those requiring hospital admission (*p* = 0.009), were also higher in the CBI group. Independent CBI predictors were BSI score (OR 3.577, 95% CI 1.233–10.378), sputum characteristics (OR 3.306, 95% CI 1.107–9.874), and radiological score (OR 1.052, 95% CI 1.004–1.102). Conclusion: According to the CBI status, two different sub-groups of patients were found on the basis of several clinical outcomes, emphasizing the importance of routine sputum microbiological monitoring. Further studies are needed to better characterize CBI profiles and to define the individual clinical impact of the most prevalent pathogenic microorganisms.

## 1. Introduction

Bronchiectasis is a respiratory disease characterised by non-reversible bronchi dilatation, usually occurring with cough, sputum production, and recurrent respiratory infections [1]. The prevalence of bronchiectasis is still unknown, although it seems to be higher than expected, being highly variable across different ages, socioeconomic status, ethnicities, and countries. Further, it is also associated with significant morbidity, leading to recurrent and longer hospital stays and worse quality of life, and even mortality [2].

Chronic bacterial infections (CBI) are often present in bronchiectasis’ patients, contributing to the maintenance of the vicious cycle of inflammation and progressive airways destruction, worst outcomes, and morbidity [1]. Therefore, these patients should be monitored with routine culture of sputum samples, in order to obtain information on possible CBI and guidance for future antibiotic therapies [3]. *Haemophilus influenzae*, *Pseudomonas aeruginosa*, *Moraxella catarrhalis*, *Streptococcus pneumoniae* and *Staphylococcus aureus* are among the most frequently isolated bacteria in bronchiectasis’ patients, although other species can also be found, such as enteric Gram-negative or nontuberculous mycobacteria [4,5,6].

It has been recognized that by using conventional microbial culture techniques, possible pathogenic microorganisms (PPM) might be isolated in a large proportion of patients, although only a restricted number of bacterial species are identified [4,5,6]. Nevertheless, new molecular techniques have shown a considerably high number and diversity of isolated PPM, even in patients with negative routine sputum cultures [7].

Interestingly, most studies published on this subject are cross-sectional rather than longitudinal, and their data are mainly based on *P. aeruginosa* infection [8,9]. In patients with bronchiectasis, few published data assessed the CBI impact on clinical and mortality outcomes [10,11,12]. Thus, nowadays, there is an insufficient knowledge on this topic.

Based on the hypothesis that CBI patients are a distinct group, this study aimed to assess the prevalence, characteristics, and risk factors for CBI in sputum samples from outpatients with bronchiectasis.

## 2. Methods

A prospective study in bronchiectasis’ patients followed in a bronchiectasis tertiary referral centre in Portugal was conducted.

Patients ≥ 18 years old, with diagnosis of non-cystic fibrosis bronchiectasis on high-resolution computed tomography (HRCT) scans and follow-up for at least one year, were enrolled in this study, with routine visits performed every six months for stable patients or three to four months for the others. The inclusion period was from February 2011 until April 2017. The local ethics committee approved this study, and informed consent was obtained from the patients.

Aetiological diagnostic work-up was made following national recommendations [13]. The post-infective aetiology was considered when the patient reported a history of a severe infection, regardless of the time between infection and the onset of bronchiectasis symptoms and excluding other aetiological causes by routine tests. Demographic data and information on aetiology, smoking history, modified Medical Research Council (mMRC) dyspnoea score, sputum characteristics (frequency, estimated 24 h volume, purulence according to Murray sputum chart [14]), Bronchiectasis Severity Index (BSI) score, lung function, and treatment at the time of the last evaluation, in a clinically stable phase, were collected. Clinical stability was defined as the absence of exacerbations and therapy changes up to four weeks prior data collection.

Spontaneous sputum samples were collected during clinical stability periods, on routine visits, and during follow-up periods. The identification of microorganisms was performed in selective media according to the Clinical and Laboratory Standards Institute [15]. Specifically, respiratory samples were homogenized and plated onto 5% sheep blood agar, chocolate agar (with a 5% CO2-enriched atmosphere), and Mac Conkey agar, and incubated for 48 h at 35 °C. Then, the isolated pathogens were identified using the proteomic technology MALDI-TOF (Biomerieux) for most of the isolated microorganisms or the biochemical panels of Vitek 2 (Biomerieux) when an alternative method was needed. All samples were evaluated for mycobacteria as well.

CBI was defined as the presence of two or more positive sputum cultures for the same PPM, at least three months apart, in a 12-month period [3]. Patients were divided into two groups according to their chronic bacterial status. Those who met CBI criteria were assigned into the CBI group, while those who did not meet all CBI criteria were allocated into the no-CBI group. Patients were considered free from chronic infection when sputum samples cultures were negative during the last 12 months.

Exacerbations were recorded at each routine visit (self-reported) and on unscheduled visits requested by the patients, due to clinical worsening. An exacerbation was registered when the patient presented a deterioration, for more than 48 h, in at least three of the following parameters: increase in cough, sputum volume and/or consistency, sputum purulence, haemoptysis/bloody sputum, breathlessness and/or exercise tolerance, fatigue and/or malaise, fever, and required antibiotics.

Lung function was determined in a clinically stable phase, according to the American Thoracic Society/European Respiratory Society guidelines [16,17]. The most recent values, expressed as absolute values (mL) and percentages of reference values, were registered and used for the analysis.

The radiological involvement degree was determined using a score based on the Bhalla scoring system [18,19]: each lobe (lingula and middle lobe were considered independent lobes) was scored according to bronchiectasis’ dilatation (0, normal; 1, <2 times; 2, 2–3 times; 3, >3 times of adjacent arteria diameter), peribronchial thickening (0, normal; 1, 20–50%; 2, >50% of bronchial lumen diameter, and 3, luminal obliteration), bronchiectasis’ extent (0, no bronchiectasis; 1, one segment; 2, at least two segments, and 3, cystic bronchiectasis), and indirect signs of small airways disease (0, no signs; 1, centrilobular nodules, tree-in-bud, bronchiolectasis, mosaic attenuation pattern, or air trapping in expiration scans). The total score varied between 0 (normal) and 60 (most severe).

### Statistical Analysis

Categorical variables were described as absolute (n) and relative frequencies, while continuous variables as mean and standard deviation (sd), or median, interquartile range (IQR), and minimum and maximum values, when appropriate.

When testing a hypothesis about continuous variables, Mann–Whitney test or t test for independent samples were used, as appropriate, considering the normality assumptions and the number of groups compared. When testing a hypothesis about categorical variables, chi-square test and Fisher’s exact test were used, as appropriate.

In order to have a more thorough understanding of CBI-associated factors, the univariate and multivariate logistic regression modeling was applied. As independent factors, in multivariate analysis, we used gender, age at diagnosis, follow-up time, BSI (with categories), sputum production, volume, radiological score, mMRC score, and exacerbation with antibiotic treatment and with hospital admission, applying a backward method. The model goodness-of-fit was assessed using the Hosmer–Lemeshow test. The significance level assumed was 0.05. Statistical analysis was performed using the Statistical Package for the Social Sciences (SPSS) software, v. 24.0.

## 3. Results

### 3.1. Patients’ Characteristics

One hundred eighty-six patients met the inclusion criteria, with a mean follow-up time of 3.8 ± 1.7 years, mean age 54.7 ± 16.2 years, and mean age at diagnosis 41.0 ± 19 years. Most patients were females (60.8%), and 22.5% were current or ex-smokers. The mean forced expiratory volume in the first second (FEV1) was 1.9 ± 0.8 L, and the mean FEV1% was 71.9 ± 27.0%.

The most common cause of bronchiectasis was infection (31.7%) followed by immune deficiencies (11.8%), whereas in 29% of cases, no cause was identified (Table 1).

Fifty-seven (30.7%) patients presented all lobes affected, and only 12 (6.5%) had only one lobe involved. The middle and right inferior lobes were the most affected regions, while the left upper lobe was the less affected. The median radiologic score was 20 (IQR 19).

### 3.2. Chronic Bacterial Infection

A total of 1746 sputum samples were analyzed, being 13 the mean number of samples per patient with CBI (maximum 42 samples), and 6 that per patient without CBI (maximum 29). There were only two patients without any sputum sample.

During follow-up, CBI was detected in 101 patients (54.3%), corresponding to a total of 136 cases of CBI, since 28 patients (27.7%) evidenced more than one different PPM (concomitant or not) causing chronic infection: 22 patients had 2 PPM, 5 had 3 and 1 had 4. Twenty different CBI-associated PPM were identified, mostly *H. influenzae* (32.3%) and *P. aeruginosa* (30.1%) (Table 2). CBI associated to *P. aeruginosa* and *H. influenzae* occurred in five patients, although never simultaneously. From the 186 patients studied, in 31 (16.7%), the same PPM was isolated at least two times without fulfilling other CBI criteria, being also *H. influenzae* (*n* = 14) the most common. Still, with regard to CBI patients, 19 (43.2%) became free from chronic infection by *H. influenzae*, 10 (24.4%) from *P. aeruginosa*, and 24 (58.5%) from the other 18 different types of bacteria.

With regards to mycobacteria isolates, in this study, three patients (1.6%) had positive cultures for *Mycobacterium tuberculosis*, and 27 (14.5%) for non-tuberculosis mycobacteria (NTM), corresponding to 67 positive cultures for NTM. *Mycobacterium avium* complex was the most frequent isolate (31.3%), followed by *Mycobacterium* species (22.4%), *Mycobacterium gordonae* (22.4%), *Mycobacterium kansassi* (8.9%), *Mycobacterium abscessus* (7.5%), *Mycobacterium chelonae* (4.5%), and *Mycobacterium peregrinum* (3.0%). Seven patients presented more than one species during the follow-up (five patients presented two different species, one patient presented three species, and one four species). Only six patients had criteria of NTM disease, according to the 2007 ATS/IDSA guidelines [20] (four patients with *M. avium* complex, one with *M.* abscessus, and one with *M. kansassi*).

### 3.3. Results according to the CBI Status

Regarding the four bronchiectasis aetiology groups shown in Table 1, differences were found between two groups (*p* = 0.021). Idiopathic bronchiectasis was the most common in CBI, while post-infection bronchiectasis was the most common in the no-CBI group. Statistically significant differences were also found between groups regarding other aspects (Table 3).

The follow-up time was higher in CBI than in no-CBI patients (4.1 vs. 3.4 years, *p* = 0.003), while the mean BSI score was worse in the CBI group (*n* = 7, moderate disease vs. *n* = 4, mild disease, *p* < 0.001). With regard to the mMRC score, significant differences (*p* < 0.003) were also found, with 23.9% of the CBI group presenting a score ≥2. Daily sputum production (77.2% vs. 56.5%, *p* = 0.002), estimated mean volume (30 mL/24 h vs. 16 mL/24 h, *p* < 0.001), and presence of purulent sputum (80.4% vs. 53.9%, *p* < 0.001) were also more prominent in patients with CBI.

The bronchiectasis’ radiological scores (total and sub-scores) obtained were worse in the CBI group (*p* < 0.001), and the number of exacerbations per year, including those requiring hospital admission, were significantly higher in the CBI group (*p* = 0.001 and 0.009, respectively). The CBI group evidenced worse lung function when compared to the no-CBI group: the median FEV1 predicted was 1.7 L versus 1.9 L, *p* = 0.008, and the residual volume was 3.1 L versus 2.7 L, *p* = 0.045. Also, the CBI group had significantly higher treatment prescription rates. No significant differences were found according to sex, age, age at diagnosis, smoking habits, body mass index (BMI), or haemoptysis.

Intending to determine the CBI predictive factors, univariate and multivariate regression analyses were performed (Table 4). Follow-up time, BSI score, mMRC score, sputum production, sputum volume and characteristics, radiological score, and exacerbations were considered as independent factors associated with CBI risk in the univariate model. All these factors were also included in the multivariate analysis, but in the final model, only BSI score, sputum characteristics, and radiological score were maintained as factors associated with higher CBI risk.

Finally, all statistical tests were repeated, excluding patients chronically infected by *P. aeruginosa*, aiming to infer the real impact of the other agents involved. Most of the significant differences persisted, except for aetiology, daily sputum production, exacerbations requiring hospitalization, lung function, long-acting muscarinic antagonists use, and inhaled antibiotic prescriptions.

During overall follow-up time, four patients died, three of which had CBI, and two underwent lung transplantation, one belonging to the CBI-group.

## 4. Discussion

Respiratory infection plays an important role in bronchiectasis, namely, on daily clinical manifestations and acute exacerbations [1,21]. The identification of microorganisms (e.g., bacteria) in respiratory samples from bronchiectasis patients is common, although the prevalence of each species varies among different studies [21]. In fact, most of these data are based on single positive sputum cultures instead of repeated samples over a period of time. Therefore, the significance of persistent bacteria isolation in clinical stable bronchiectasis’ patients is not totally clear.

Despite sputum samples could be contaminated by the upper airway microbiome [22], they constitute the best way to obtain culture samples in a longitudinal study, given that their retrieval is a non-invasive, easily accessible and inexpensive procedure. Many studies have been done on *P. aeruginosa* colonisation infection prevalence and its importance [6,9], but little is known on other bacteria.

In our series, more than half (54.3%) of the patients presented CBI criteria, with 20 different bacteria causing chronic infection. The most common bacteria were *H. influenzae* (32.3%) and *P. aeruginosa* (30.1%). Our findings are similar to those previously reported in two multicentre studies (50% and 40%, respectively), despite the fact that both *H. influenzae* and *P. aeruginosa* rates were much lower (22, 16% and 15, 15%, respectively) than those observed here [10,11]. In our centre, the prevalence of *Pseudomonas* infection (30.1%) stands between the value described in Spain (21.2%) and Greece (36.5%) [10]. So, our higher rates may reflect distinct bacterial epidemiology and antibiotic strategies, longer follow-up periods, and the fact that many patients were already chronically infected when referred to our Bronchiectasis Centre.

When considering the cases with the same PPMs, at least in two occasions and without fulfilling all criteria of chronic infection, *H. influenzae* was also the most common. This fact confirms the high prevalence of this PPM in bronchiectasis’ patients [21], although culture-independent techniques have shown that *H. influenzae* is under-recognised by culture [7,23]. Beyond that, more than one pathogen associated with chronic infection was found in the same patient in 27.7% of cases, simultaneously or during follow-up. In fact, McDonnell et al. [6] also found a similar polymicrobial rate (25.2%), suggesting a dynamic, complex, and polymicrobial infection, with changes occurring over time. More recently, Cox et al. [23], on the basis of new data from sputum molecular analysis, verified the occurrence of frequent mixed infections, often containing multiple pathogens.

The number of cases that became CBI-free was lower for *P. aeruginosa* than for other bacteria, because of this bacterium’s great ability to bronchiectatic airways adaptation and persistence [21]. However, being *P. aeruginosa*-free, did not mean for a patient that eradication was truly achieved, as the bacterial load might be too small to be detected by sputum culture.

In our study, CBI were also significantly more common in idiopathic bronchiectasis. This association has not been previously documented [11,24] and could rise the hypothesis that part of these cases were due to a dysregulated inflammatory response to airway bacterial infection [25].

CBI patients showed worse disease outcomes, such as higher BSI, mMRC dyspnoea score, sputum frequency, volume and purulence, radiological score, exacerbations rate, as well as worst lung function. They also had a longer follow-up time, perhaps reflecting a more symptomatic disease, being referred earlier to the Bronchiectasis Clinic. It is also interesting to point out that some of these risks were already identified in stable chronic obstructive pulmonary disease (COPD) patients with chronic bronchial infection, including FEV1 reduction, increased exacerbations rate, and worse health status [26]. These findings make us believe that persistent infection is a factor, in itself, relevant, with similar effects regardless of the underlying disease.

On the other hand, the definition of pulmonary exacerbation used in this study was established at the beginning of patients’ inclusion, being almost the same as the recent definition published by the EMBARC/BRR working group [27]. It only differs in two criteria: fever and decision to start antibiotics instead of any treatment change. The CBI group evidenced a significantly higher exacerbations rate, including severe episodes requiring hospital admission.

Multivariate logistic regression analysis identified three independent factors for CBI: BSI score, sputum characteristics (mucopurulent), and radiological score. Other studies have already demonstrated a correlation between earlier bronchiectasis diagnosis and risk for CBI and purulent sputum production [14,22]. This probably reflects a more severe disease status with early symptoms and might also justify the higher sputum purulence. The BSI score comprises a bacterial chronic colonisation item, emphasising the importance of chronic infection status in bronchiectasis patients [12]. In our study, the BSI score was the most relevant independent factor. This finding highlights the importance that knowledge of chronic infection status may have in these patients.

When chronic infection by *P. aeruginosa* was excluded, most of statistically significant differences remained, which underlines the contribution of the other chronic bacteria to the CBI results, when studied as a single group. However, it should be highlighted that the no-*Pseudomonas* CBI group was not associated with significant differences on important outcomes, such as lung function or exacerbation requiring hospitalization. Thus, our study points to an association between CBI status and a more severe group of patients, according with recent studies [10,11,12]. In a cluster analysis of 1145 bronchiectasis patients from five European centres, four clusters were identified, based on the presence and type of chronic infection: chronic infection with *P. aeruginosa* (16%), other chronic infection (24%), patients with daily sputum without chronic infection, and patients without daily sputum and without chronic infection (59%). This study confirmed that the bacterial infection status is sufficient to define clinical phenotypes. However, the authors’ findings differed in terms of clinical characteristics, radiological score, BSI, FEV1%, quality of life, exacerbations, hospitalisations due to exacerbations, and inflammatory profile, from the less severe without chronic infection to the most severe, when chronic infection by *P. aeruginosa* is present. As in previous studies, in our series, patients chronically infected by *P. aeruginosa* and other PPMs corresponded to a non-negligible percentage of the entire patient population (22% and 32%, respectively).

Several studies have shown that chronic infection constitutes a local [28] and systemic [29] inflammatory trigger [29,30], contributing to the vicious cycle of inflammation, infection, and disease progression. However, it is not totally clear if CBI predisposes to disease progression or if it is only a severity marker. Even in the case of chronic *Pseudomonas* infection, on which there are more studies, this doubt still persists [31], and it is still to be clarified what predisposes some patients to be chronically infected. Thus, it can be assumed that, by using new nonculture-based techniques during follow-up, more cases of CBI would be detected, given the great sensitivity of these methodologies [7]. Nevertheless, taking into account the differences stated between the two analyzed groups, it is possible to assume that repeated sputum cultures, despite their limitations, continue to be a simple and reliable method to identify chronically infected patients and, therefore, should be part of routine clinical practice [3].

As the main strengths of this study, we underline that this is the first study performed in Portugal on this topic, it presents a high uniformity in therapeutic interventions, given that it was performed in only a single referral centre, analysed a considerable number of samples, and measured follow-up periods. On the other hand, the study limitations include the number of patients (small sample size, heterogeneity)—with a low number of cases within each CBI group, which did not allow an individual analysis of the impact of different PPMs—and the absence of a pre-stablished period between the collections of sputum samples. Also, we did not use questionnaires to measure the patients’ quality of life nor associated comorbidities scores, which could have helped to better distinguish the two groups of patients.

## 5. Conclusions

Overall, this study highlights the diversity of the distal airway microbiome contributing to CBI, its impact on several clinical endpoints, and the importance of routine sputum microbiological evaluation to define the status of chronic infection. Larger, multicentre, and longitudinal studies are needed to better characterize CBI profiles and to define the individual clinical impact of the most prevalent PPMs. Comparing the clinical evolution before and after the onset of the CBI status could also be very helpful. This knowledge may lead to higher evidence-based treatment strategies for the eradication and long-term antibiotic treatment of other bacteria besides *P. aeruginosa*.

## Figures and Tables

**Table 1 jcm-08-00315-t001:** Bronchiectasis aetiology. CBI, chronic bacterial infections.

Aetiology	CBI	No CBI	All Patients
**Post-infective ^α^**	29 (28.7)	30 (35.3)	59 (31.7)
**Idiopathic**	38 (37.6)	16 (18.8)	54 (29.0)
**Immunodeficiency ^β^**	9 (8.9)	13 (15.3)	22 (11.8)
**Others**	25 (25.0)	26 (30.6)	51 (27.4)
Asthma	5 (5.0)	6 (7.1)	11 (5.9)
Auto-immune inflammatory disease	3 (3.0)	6 (7.1)	9 (4.8)
Primary ciliary dyskinesia	8 (7.9)	1 (1.2)	9 (4.8)
Swyer James MacLeod S.	0	5 (5.9)	5 (2.7)
COPD	2 (2.0)	2 (2.4)	4 (2.2)
Inflammatory bowel disease	1 (1.0)	4 (4.7)	4 (4.7)
A1AT deficiency	3 (3.0)	0	3 (1.6)
ABPA	1 (1.0)	1 (1.2)	2 (1.1)
CFTR related disease ^χ^	2 (2.0)	0	2 (1.1)
Gastric reflux	0	1 (1.2)	1 (0.5)
**Total**	101 (100)	85 (100)	186 (100)

^α^ Post-pneumonia (*n* = 33), post-tuberculosis (*n* = 22), and others (*n* = 4); ^β^ Common variable immunodeficiency (*n* = 6), followed by IgA deficiency (*n* = 4), agammaglobulinemia (*n* = 3), hyper IgM syndromes (*n* = 3), and others (*n* = 6); ^χ^ Borderline sweat tests (*n* = 2), extensive sequencing of the *CFTR* gene: IVS8-5T (*n* = 1), C.1727G>C(p.Gly576Ala)/C.2002C>T(p.Arg688Cys). A1AT, Alpha1 anti-trypsin; ABPA, allergic bronchopulmonary aspergillosis; CFTR, cystic fibrosis transmembrane conductance regulator; COPD, chronic obstructive pulmonary disease; Swyer James MacLeod S, Swyer James MacLeod Syndrome.

**Table 2 jcm-08-00315-t002:** CBI-associated bacteria.

Isolated Bacteria	CBI	All Samples
*Haemophilus influenzae*	44 (32.3)	174
*Pseudomonas aeruginosa*	41 (30.1)	333
*Stenotrophomonas maltophilia*	8 (5.9)	59
Meticilin-sensitive *Staphylococcus aureus*	6 (4.4)	27
*Serratia marcescens*	5 (3.6)	20
*Streptococcus pneumoniae*	5 (3.6)	18
*Moraxella catarrhalis*	5 (3.6)	18
*Achromobacter* spp.	4 (2.9)	15
Others (12 different PPM) *	23 (16.9)	18
**Total**	136	664

PPM, possible pathogenic microorganisms. * Others, 12 PPMs identified, included *Enterobacter cloacae*, *Enterobacter aerogenes*, *Klebsiella oxytoca*, *Klebsiella pneumoniae*, *Escherichia coli*, *Citrobacter freudii*, *Proteus mirabilis*, *Raoutella planticola*, Multiple-Resistant *Staphylococcus aureus* (*SAMR*), *Morganela morganii*, *Pseudomonas stutzeri*, and *Nocardia* spp.

**Table 3 jcm-08-00315-t003:** Patient characteristics and differences between groups.

Characteristics	General Sample			CBI Excluding *P. aeruginosa*		
	CBI (*n* = 101)	No CBI (*n* = 85)	*p* Value	CBI (*n* = 60)	No CBI (*n* = 85)	*P* Value
**Female sex (%)**	62.4	58.8	0.621	45.7	58.8	0.169
**Age at diagnosis (m ± sd)**	40 ± 20	41 ± 18	0.532	38 ± 19	41 ± 18	0.207
**Age at last evaluation (m ± sd)**	56.4 ± 15.9	52.8 ± 16.4	0.237	53.2 ± 16.6	52.8 ± 16.4	0.958
**Follow-up time years, median**	4.1	3.4	**0.003**	4.2	3.4	**0.037**
**Smoking habits, *n* (%)**			0.118			0.477
Smoker	2 (2.0)	7 (8.2)		2 (3.3)	7 (8.2)	
Ex-smoker	20 (19.8)	13 (15.3)		10 (16.7)	13 (15.3)	
Never smoker	79 (78.2)	65 (76.5)		48 (80.0)	65 (76.5)	
**BMI (m ± sd)**	24.5 ± 4.0	25.5 ± 4.6	0.132	24.5 ± 4.3	25.5 ± 4.6	0.164
**BSI (m ± sd)**	7 ± 4.0	4 ± 2.0	**<0.001**	5 ± 4.0	4 ± 2.0	**0.003**
**mMRC score, *n* (%)**			**0.003**			**0.011**
0	48 (47.5)	39 (45.9)		33 (55)	39 (45.9)	
1	29 (28.7)	41 (48.2)		18 (30)	41 (48.2)	
2	14 (13.9)	3 (3.59)		6 (10)	3 (3.59)	
3	5 (5.0)	2 (2.4)		0	2 (2.4)	
4	5 (5.0)	0		3 (5)	0	
**Sputum production, *n* (%)**			**0.002**			0.118
Daily	78 (77.2)	48 (56.5)		41 (68.3)	48 (56.5)	
Not-daily	15 (14.9)	15 (17.6)		12 (20.0)	15 (17.6)	
Only in exacerbations	8 (7.9)	22 (22.9)		7 (11.7)	22 (22.9)	
**Volume (mL/24 h), (m ± sd)**	30 ± 28.0	16 ± 11.0	**<0.001**	26 ± 22.0	16 ± 11.0	**0.017**
**Sputum characteristics, *n* (%)**			**<0.001**			**0.001**
Mucoid	18 (19.6)	29 (46)		9 (17.0)	29 (46)	
Mucopurulent	52 (56.5)	30 (47.6)		30 (56.6)	30 (47.6)	
Purulent	18 (19.6)	4 (6.3)		12 (22.6)	4 (6.3)	
Very purulent	4 (4.3)	0		2 (3.8)	0	
**Haemoptysis (%)**	42.9	32.9	0.178			
**Exacerbations/year**						
Antibiotic treatment (median; IQR)	0.85 (1.3)	0.51 (1.0)	**0.001**	0.84 (1.2)	0.51 (1.0)	**0.005**
Hospitalization (median; min-max)	0 (0–0.9)	0 (0–2.2)	**0.009**	0 (0–1.2)	0 (0–2.2)	0.488
**Radiological score, median (IQR)**	24 (19.0)	15 (17.0)	**<0.001**	21 (19.0)	15 (17.0)	**0.015**
Bronchiectasis dilatation	6 (7.0)	4 (5.0)	**<0.001**	6 (6.0)	4 (5.0)	**0.025**
Peribronchial thickening	4 (5.0)	2 (3.0)	**0.001**	3 (5.0)	2 (3.0)	0.055
Bronchiectasis extension	9 (11.0)	6 (7.0)	**<0.001**	8 (7.0)	6 (7.0)	**0.034**
Indirect signs small airways disease	3 (4.0)	2 (3.0)	**<0.001**	2 (3.0)	2 (3.0)	0.094
**Lung function, median (IQR)**						
FVC L	2.6 (1.2)	2.8 (1.3)	0.059	2.7 (1.1)	2.8 (1.3)	0.439
FVC %	87.0 (29)	92.6 (29.6)	0.112	92.0 (23.2)	92.6 (29.6)	0.967
FEV1 L	1.7 (1.1)	1.9 (1.3)	**0.008**	1.8 (1.0)	1.9 (1.3)	0.332
FEV1 %	68.5 (36.7)	74.6 (38.1)	0.062	1.8 (29.9)	74.6 (38.1)	0.883
RV L	3.1 (1.1)	2.7 (0.9)	**0.045**	2.8 (1.02)	2.7 (0.9)	0.686
RV %	162.0 (62.3)	150.5 (38.3)	0.163	155.9 (57.6)	150.5 (38.3)	0.403
RV/TLC L	52.0 (14.6)	48.4 (17.1)	**0.004**	47.9 (13.4)	48.4 (17.1)	1.149
RV/TLC %	143.0 (40.4)	136.3 (33.9)	0.103	142.6 (39.5)	136.3 (33.9)	0.376
**Treatment (%)**						
Inhaled bronchodilators						
LAMA	88.1	78.8	**0.008**	52.5	78.8	0.287
LABA	64.0	43.5	**0.005**	81.7	43.5	0.673
Mucolytic therapy	73.7	26.3	**0.002**	40.0	26.3	**0.010**
Hypertonic saline	10.9	0	**0.002**	6.7	0	**0.028**
Inhaled antibiotics	9.9	0	**0.002**	3.3	0	0.170
Macrolides	34.7	1.2	**0.001**	23.3	1.2	**<0.001**
Adherence to airway clearance	64.8	35.2	**0.001**	68.3	35.2	**0.003**

^β^ according to Murray scale. BMI, body mass index; BSI, bronchiectasis severity index; FEV1, forced expiratory volume in 1 s; FVC, forced vital capacity; LABA, long-acting β2-agonists; LAMA, long-acting muscarinic antagonist; mMRC, modified Medical Research Council; RV, residual volume; TLC, total lung capacity. Bold: *p* < 0.05.

**Table 4 jcm-08-00315-t004:** Logistic regression analysis of CBI predictive factors.

Factor	Univariate Analysis			Multivariate Analysis ^1^		
	OR	IC 95%	*P* Value	OR	IC 95%	*P* Value
**Follow-up time, y**	**1.001**	**1.000–1.001**	**0.004**			
**BSI**	**1.423**	**1.248–1.623**	**<0.001**			
<4	Ref		**<0.001**	Ref		
4–8	**3.527**	**1.797–6.923**	**<0.001**	**3.577**	**1.233–10.378**	**0.019**
≥9	**12.086**	**4.088–35.734**	**<0.001**	**7.611**	**1.221–47.455**	**0.030**
**mMRC score**			**0.002**			
0	Ref					
1	0.575	0.304–1.086	0.088			
≥2	**3.792**	**1.016–14.145**	**0.011**			
**Sputum production**			**0.003**			
Daily	Ref			Ref		
Not daily	0.615	0.276–1.371	0.235			
Only in exacerbations	**0.224**	**0.092–0.542**	**0.001**			
**Sputum volume**	**1.056**	**1.025–1.089**	**<0.001**	1.035	0.996–1.076	0.079
**Sputum characteristics**			**0.001**			
Mucoid	Ref			Ref		
Mucopurulent	**2.793**	**1.332–5.854**	**0.007**	**3.306**	**1.107–9.874**	**0.032**
Purulent + Very purulent	**8.861**	**2.624–29.921**	**<0.001**	2.063	0.348–12.216	0.425
**Radiological score**	**1.055**	**1.027–1.084**	**<0.001**	**1.052**	**1.004–1.102**	**0.034**
Bronchiectasis dilatation	**1.123**	**1.044–1.208**	**0.002**			
Peribronchial thickening	**1.198**	**1.082–1.327**	**0.001**			
Bronchiectasis extension	**1.126**	**1.052–1.206**	**0.001**			
Indirect signs small airways disease	**1.328**	**1.140–1.546**	**<0.001**			
**Exacerbations/year**						
Exacerbation with antibiotic	**1.935**	**1.321–2.833**	**0.001**			
Exacerbation with hospital admission	**5.395**	**1.200–24.266**	**0.028**			
**FEV1 %**						
<50%	Ref					
≥50%	0.808	0.11–5.864	0.833			

^1^ Independent variables: gender, age at diagnosis, follow-up time, BSI (with categories), sputum production and volume, radiological score (scale); FEV1, forced expiratory volume in 1 s; mMRC score, exacerbation with antibiotic, exacerbation with/without hospital admission. Hosmer and Lemeshow, *p* = 0.580. Bold: *p* < 0.05.

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
