# Peer review of "Chronic Bacterial Infection Prevalence, Risk Factors, and Characteristics: A Bronchiectasis Population-Based Prospective Study"

_jcm, 2019, doi:10.3390/jcm8030315_

Reviewer 1 Report

'This descriptive longitudinal study aims to characterise the bacterial organisms present in sputum during periods of clincial stability and exacerbations in patients with bronchiectasis .It adds to and confirms current literature findings. 

MInor points

1. How regularly were routine visits conducted  ? could you please clarify 

2. What percentage of samples cultured TB or NonTuberculous Mycobacteria . It woudl be useful and contextual to describe this even though further work was not undertaken with these  data. 

Author Response

This descriptive longitudinal study aims to characterize the bacterial organisms present in sputum during periods of clinical stability and exacerbations in patients with bronchiectasis. It adds to and confirms current literature findings.

Answer: The authors thank the overall appreciation of our work.

Minor points

1. How regularly were routine visits conducted? could you please clarify 

Answer: Thanks for the reviewer question. The visits were performed in stable patients every 6 months, and in the other patients, on average, every 3-4 months. This information was duly included on lines 70-71.

2. What percentage of samples cultured TB or Non-Tuberculous Mycobacteria. It would be useful and contextual to describe this even though further work was not undertaken with these data.

Answer: Thank you for the reviewer question. Given the small number of articles evaluating the impact of mycobacteria on non-CF bronchiectasis as well as the reduced number of patients with NTM-positive cultures, and the extent of data already analyzed in this article, we decided to analyze mycobacterial infection in an individualized work with longer follow-up and with more patients. Some results on this aspect were carefully added in lines 169-176.

Reviewer 2 Report

This is a retrospective audit of Portuguese patients with non CF bronchiectasis (NCB).  A total of 186 patients attending a single tertiary referral centre were included.

The authors have found phenotypic differences between patients defined as having chronic bacterial infection (CBI)  ie  >= 2 positive sputum cultures for the same potentially pathogenic microorganism (PPM) at least 3 months apart in a 12 month period. 

In the CBI group patients were sicker, with more sputum, more exacerbations needing antibiotics and more hospital admissions.  This is an important if unsurprising  finding.

Some issues:

A.

An important detail missing from the manuscript and critical for the understanding of these data, is the frequency of sputum cultures.  The mean number of samples in CBI patients was 13.  The mean for non CBI was 6 over a mean follow up time of 3.8 years

So what was the mean number of samples per year in the CBI vs non CBI group?  Were the non CBI defined by fewer samples being done?  Or alternately,  what was the mean period of time between sputum cultures – was it longer in the non CBI group.  After all a diagnosis of CBI cannot be made if the sputum is not analysed. Maybe the non CBI group just didn’t have enough samples done?

The finding of apparently real differences between the CBI and non CBI groups is against this possibility but it needs to be clarified.

B.

The aetiology of bronchiectasis is listed.  None of these are defined. How was “Post infective” bronchiectasis diagnosed?.  What does this term mean.  31`% of patients fell in this category.  Is it childhood infection? Mean age of diagnosis was 40 years which would seem to exclude a childhood onset.  Similarly “Immunodeficiency” should be clarified.   Also did the patients with “CFTR related disease” have mild CF related bronchiectasis with a second allele that was undefined?

 In table 3 the data runs over 2 pages.  The header needs to be on the top of the second page as well.

The absence of data on NTM infection (likely to be significant) is disappointing.  The authors hint this will be discussed in another paper but it is part of a discussion of “chronic bacterial infection”.

As the authors state whether the regular presence of bacteria is the cause or consequence of the severity of the NCB remains uncertain.  The association remains however and may be hypothesis generating for future interventional studies.

Author Response

This is a retrospective audit of Portuguese patients with non-CF bronchiectasis (NCB). A total of 186 patients attending a single tertiary referral centre were included. The authors have found phenotypic differences between patients defined as having chronic bacterial infection (CBI), ie >= 2 positive sputum cultures for the same potentially pathogenic microorganism (PPM) at least 3 months apart in a 12-month period. In the CBI group patients were sicker, with more sputum, more exacerbations needing antibiotics and more hospital admissions. This is an important if unsurprising finding.

Answer: The authors thank the overall appreciation of our work.

Some issues:

A.

An important detail missing from the manuscript and critical for the understanding of these data, is the frequency of sputum cultures. The mean number of samples in CBI patients was 13. The mean for non-CBI was 6 over a mean follow-up time of 3.8 years.

So, what was the mean number of samples per year in the CBI vs non-CBI group? Were the non-CBI defined by fewer samples being done?  Or alternately, what was the mean period of time between sputum cultures – was it longer in the non-CBI group. After all a diagnosis of CBI cannot be made if the sputum is not analyzed. Maybe the non-CBI group just didn’t have enough samples done? The finding of apparently real differences between the CBI and non-CBI groups is against this possibility but it needs to be clarified.

Answer: Thanks for the reviewer question. In these patients, sputum cultures are requested at each visit, averaged every 6 months in stable patients and 3-4 months in other patients (this information was added in lines 70-71). As we see in clinical practice, more severe patients are observed more frequently, which also contributes to the greater number of samples collected. On the other hand, the production of sputum is not similar among the patients, some of them presenting sputum production only during the period of exacerbation or non-daily, thus limiting the frequency of the sputum samples collection. According to the results obtained in this study, both the pattern and volume of sputum production in CBI patients were statistically higher when compared to non-CBI patients. Both reasons may justify the differences found between the number of samples obtained in CBI vs non-CBI groups. Thus, the only way to standardize the collection of sputum samples would be the programming of spontaneous or induced sputum at pre-established periods of time. This aspect is of paramount importance and, therefore, was considered a limitation of our work and added in the main manuscript (see lines 313-314).

B.

The etiology of bronchiectasis is listed.  None of these are defined. How was “Post infective” bronchiectasis diagnosed? What does this term mean. 31% of patients fell in this category. Is it childhood infection? Mean age of diagnosis was 40 years which would seem to exclude a childhood onset. Similarly, “Immunodeficiency” should be clarified. Also did the patients with “CFTR related disease” have mild CF related bronchiectasis with a second allele that was undefined?

Answer: Thank you for the reviewer question. Regarding the etiology of “Immunodeficiency”, this information was carefully introduced in Table 1 (lines 156-159). With regards to “Post infective” diagnosis, given the huge variability and absence of a consensual definition, we included the definition considered in the methodological section (lines 74-77) and the specific infections found in Table 1 (l. 156-159). The CFTR-related disease was considered in 2 patients who did not fulfil the CF criteria, but both had bronchiectasis, chronic pancreatitis, with other etiologies excluded, sweat tests border line, where the extensive CFTR gene sequencing, led to the identification of IVS8-5T allele in one patient and in another one 2 mutations were identified, but with uncertain clinical significance (l. 158-159).

In table 3 the data runs over 2 pages. The header needs to be on the top of the second page as well.

Answer: Thank you for the reviewer advice. This aspect was revised in Table 3.

The absence of data on NTM infection (likely to be significant) is disappointing. The authors hint this will be discussed in another paper but it is part of a discussion of “chronic bacterial infection”. 

Answer: Thank you for the reviewer question. Given the small number of articles evaluating the impact of mycobacteria on non-CF bronchiectasis as well as the reduced number of patients with NTM-positive cultures, and the extent of data already analyzed in this article, we decided to analyze mycobacterial infection in an individualized work with longer follow-up and with more patients. Some results on this aspect were carefully added in lines 169-176.

As the authors state whether the regular presence of bacteria is the cause or consequence of the severity of the NCB remains uncertain. The association remains however and may be hypothesis generating for future interventional studies.

Answer: Thank you for the reviewer opinion. We completely agree with the reviewer perspective. Indeed, until now there is a scarce number of studies evaluating the impact of CBI on both severity and prognosis of NCB, excepting for Pseudomonas aeruginosa, and is even inexistent studies assessing the causal relationship between infection and disease severity. As we mentioned in the conclusion section, one way to overcome this uncertainty would be to compare the severity of the disease before and after acquiring a chronic infection. On the other hand, a second option would be the development of interventional studies with distinct therapeutic strategies (e.g. comparing eradication vs. non-eradication strategies for other microorganisms besides P. aeruginosa) toward to define the best clinical approach.